



# Development and validation of satellite-derived surface NO$_2$ estimates using machine learning versus traditional approaches in North America

Debora Griffin[1], Colin Hempel[1,2], Chris McLinden[1,3], Shailesh Kumar Kharol[4], Colin Lee[1], Andre Fogal[1,5], Christopher Sioris[1], Mark Shephard[1], and Yuan You[1]

[1]Air Quality Research Division, Environment and Climate Change Canada, Toronto, Ontario, Canada
[3]Department of Physics, University of British Columbia, Vancouver, Canada
[3]Department of Physics and Engineering Physics, University of Saskatchewan, Saskatoon, Saskatchewan, Canada
[4]AtmoAnalytics Inc., Brampton, Canada
[5]Department of Physics and Engineering Physics, University of Waterloo, Waterloo, Canada

**Correspondence:** Debora Griffin (debora.griffin@ec.gc.ca)

**Abstract.** Nitrogen dioxide (NO$_2$) is one of the key pollutants with profound implications for air quality, and human health, and is needed to establish the air quality health index (AQHI). Currently, over 600 surface air monitoring stations are distributed across Canada and the United States measuring NO$_2$, but many areas remain unmonitored leading to incomplete information for health risk assessments. This study leverages Tropospheric Monitoring Instrument (TROPOMI) satellite observations and

machine learning models to derive high-resolution surface NO$_2$ concentrations, provides enhanced spatial coverage and accuracy, revealing urban-rural NO$_2$ gradients across North America. Existing traditional methods rely on scaling with modeled profiles to obtain NO$_2$ surface concentrations from satellite observations. Here, we compare this traditional method to a machine learning approach that utilizes NO$_2$ observations from TROPOMI, together with meteorological parameters, land cover type, topography, and emission inventories. Our results show that the machine learning (using random forest) yields less bias

between the surface monitoring measurements and the "satellite-derived" surface concentrations, significantly improved the correlation coefficient (R$^2$ ∼0.77-0.91) compared to the traditional method (R$^2$ ∼0.39-0.57) and yields to significantly less bias.

## 1 Introduction

Nitrogendioxide (NO$_2$) is a highly reactive gas that is a major component of outdoor air pollution. It is primarily produced

by combustion processes, including burning fossil fuels in motor vehicles, power plants, and industrial processes. NO$_2$ can also be formed through natural processes such as lightning and microbial activity in soils. NO$_2$ plays a significant role in the production of tropospheric ozone and has adverse effects on the environment and human health. Exposure to high levels of NO$_2$ can cause a range of adverse health effects, particularly on the respiratory system Health Canada (2024); Environment Canada (2024). As such it is often included in the calculation of air quality indices, a metric which communicates the health



risk associated with pollution levels in ambient air. Some examples include the US Air Quality Index, the Air Quality Health Index in Canada, the European Air Quality Index, and the National Air Quality Index in India.

$NO_2$ is one of several air pollutants regulated by national and international air quality standards, and efforts to reduce $NO_2$ emissions from transportation and industry are an important part of air quality management strategies. Satellite and ground-based measurements have shown significant progress in the reduction of $NO_2$ emissions, including across the US and Canada

Russell et al. (2012); Kharol et al. (2015). Surface concentrations are monitored in Canada through the National Air Pollution Surveillance Program (NAPS; https://www.canada.ca/en/environment-climate-change/services/air-pollution/monitoring-networks-data/national-air-pollution-program.html), with the mission to provide accurate and long-term air quality data across Canada. Currently, there are roughly 290 NAPS $NO_2$ monitoring stations. The equivalent in the US is the Air Quality System (AQS) operated by the Environmental Protection Agency (EPA) (https://www.epa.gov/outdoor-air-quality-data), which has ap-

proximately 450 $NO_2$ monitoring stations at present. Even with several hundred monitoring stations across Canada and the US, there remains wide monitoring gaps, especially in more remote areas and smaller communities.

Satellite observations can help to fill some of these gaps with previous studies showing the capability of satellite observations to detect ground-level $NO_2$ Goldberg et al. (2021); Jeong and Hong (2021). However, these satellite-based sensors observe vertical column densities (VCDs) rather than just the near-surface concentrations. but rather $NO_2$ through the entire atmospheric

column. VCDs from the Tropospheric Monitoring Instrument (TROPOMI), a satellite sensor show a strong correlation with surface concentrations, indicating some sensitivity near the surface, which allows for the inference of surface concentrations Goldberg et al. (2021); Jeong and Hong (2021). There are several ways to infer surface concentrations from satellite observations. Traditionally, model scaling is a common approach where a ratio between model surface and model VCDs is applied to the satellite VCDs Lamsal et al. (2008); McLinden et al. (2014); Kharol et al. (2015); Griffin et al. (2019); Cooper et al.

(2020). This heavily relies on the accuracy of the air quality model or chemical transport (CTM) model and the winds that drive the model, and can introduce errors when the location of the emissions or the wind direction and speed used by the CTM model is not correct. More recently, machine learning has become more popular with the advancement of new technologies. When used with caution, machine learning is a powerful tool that can analyze large datasets and identify patterns that are not easily recognizable by humans. There have been several studies on using machine learning (ML) algorithms with TROPOMI

$NO_2$ troposheric columns to generate surface $NO_2$ concentrations in different parts of the world, including China Long et al. (2022); Grzybowski et al. (2023), and Germany Chan et al. (2021). Further information specifically on studies using machine learning to obtain surface $NO_2$ from satellite observations can be found in Siddique et al. Siddique et al. (2024). Currently, to our knowledge, there is little machine learning done to derive surface concentrations in less populated areas such as Canada.

The goal of this work is to develop a ML-based surface $NO_2$ product that is reliable for both near-real time monitoring

as well as for retrospective environmental and health impact studies. Here, we compare our machine learning with the traditional method of obtaining surface concentrations from satellite observations in North America, and highlight some of the relevant challenges. Our dataset of $NO_2$ surface concentrations is publicly available to download https://hpfx.collab.science.gc.ca/~deg001/surfaceNO2.





## 2    Datasets and Methodology

### 2.1    TROPOMI NO$_2$

TROPOMI (TROPOspheric Monitoring Instrument) is a satellite instrument designed to observe the nitrogen dioxide (NO$_2$) in the troposphere Hu et al. (2018); Veefkind et al. (2012). It is part of the European Space Agency's (ESA) Sentinel-5 Precursor mission, which aims to provide accurate and reliable atmospheric composition information for air quality and climate change. TROPOMI is a hyperspectral imaging spectrometer that operates in the ultraviolet, visible, near-infrared, and shortwave infrared spectral regions. It uses a push-broom scanning technique to capture high spatial-resolution images of the Earth's atmosphere in the UV-visible with a ground resolution of up to 3.5 km x 5.5 km (since August 2019, 7km x 5.5 km prior). This provides the possibility of detecting and estimating NO$_2$ emissions, including urban and industrial regions Griffin et al. (2019); Goldberg et al. (2019, 2024), shipping lanes Riess et al. (2022), and power plants Beirle et al. (2019); Dix et al. (2022). In this study we use version 2 of the NO$_2$ TROPOMI dataset (v2) and remove observations that have a quality flag below 0.75. Additionally, in our final estimated NO$_2$ surface concentrations we also remove points over water as we do not have any training data (surface monitors) over water.

### 2.2    Air Quality Monitoring Stations

US and Canadian NO$_2$ from surface monitoring stations are used for two purposes: a portion is used to train the ML system while the remainder is used to evaluate the ML product.

The U.S. Environmental Protection Agency (EPA) has established a national network of air quality monitoring stations to measure various air pollutants, including NO$_2$. These monitoring stations are part of the EPA's Air Quality System (AQS) and provide hourly measurements of NO$_2$ concentrations at the surface level. The EPA's NO$_2$ monitoring network consists of approximately 450 sites in the United States. Similarly, in Canada, the National Air Pollution Surveillance (NAPS) program is a national network of air quality monitoring stations. The NAPS program has established a network of approximately 290 air quality monitoring stations across Canada.

The NO$_2$ measurements obtained by the AQS and NAPS networks are collected using chemiluminescence-based analyzers, which measure the concentration of NO$_2$ in the ambient air. These instruments operate alternately in nitric oxide (NO) and NO$_x$ mode, and the NO$_2$ concentrations are inferred indirectly as the difference between measurements obtained in the NO$_x$ mode and NO mode. Due to interference from other reactive nitrogen species, such as peroxyacetyl nitrate (PAN), nitric acid (HNO$_3$), nitrous acid (HONO), and organic nitrates, the NO$_2$ concentrations can be overestimated by the chemiluminescence-based analyzers Winer et al. (1974); Demerjian (2000); Steinbacher et al. (2007). Previous studies have accounted for this by applying a correction using modeled PAN, HNO$_3$, and alkyl nitrates Lamsal et al. (2008, 2010); Kharol et al. (2015). In recent years, there have been advancements made to improve the accuracy of the NO$_2$ concentrations measured with these instruments, including selective catalytic reduction, scrubbers, and improved calibration. Thus, we have not applied a correction term to the NAPS and AQS NO$_2$ dataset.





To establish the training dataset, we calculate the 12:00–15:00 average – a time range representative of the TROPOMI overpass – for all the in situ $NO_2$ measurements over US and Canada for the time period between 2018 and 2022 (except in Sect. 3.1 we use 2018-2021 for training, as 2022 is used for validation). The location of the stations is shown in Figure 1 with AQS as blue (450 sites) and NAPS (289 sites) as red points. This map also shows the locations of the model points (grey points, 290 sites) used (as further discussed in Sect. 3.2).

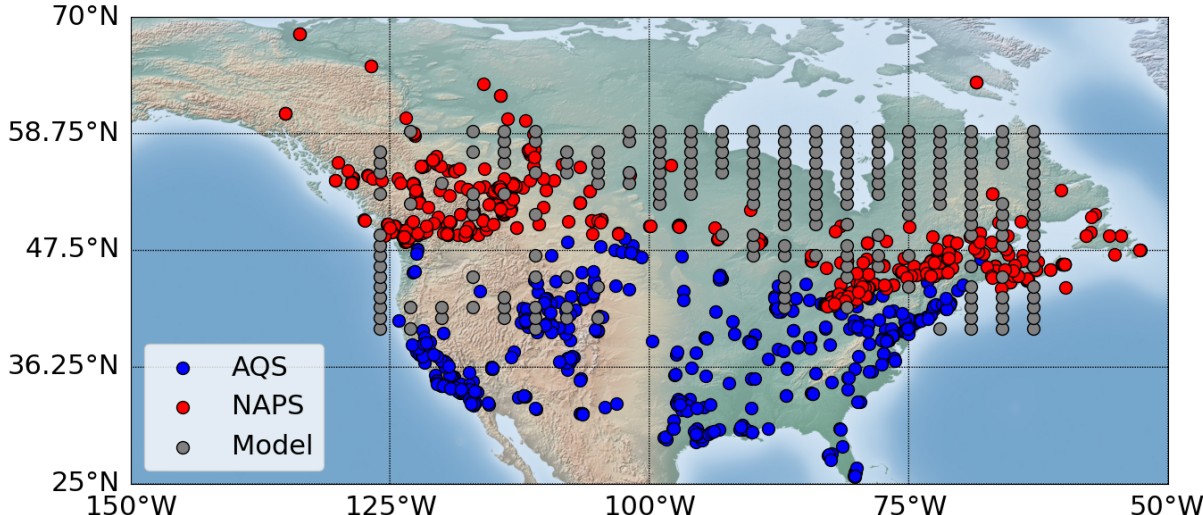

**Figure 1.** Location of the AQS (blue points) and NAPS (red points) stations that provide $NO_2$ surface concentration measurements. Also shown is the location of additional model output used for the training of the random forest algorithm (grey points).

90

## 2.3 Machine Learning and Training methods

To predict surface level $NO_2$, we utilize the machine learning algorithms in python sklearn. As part of this study, we tested several machine learning methods, including neural networks, decision tree, and random forest. In this study, we found that the random forest led to the best results by far, leading to correlations around $R^2 \sim 0.8$ between estimated daily mid-day and

95    measured $NO_2$ surface concentrations (see Sect. 3).

The random forest machine learning algorithm consists of a random collection of decision trees and is trained on random subsets of features and data. The strength of this method is its robustness, simple implementation and its accountability for non-linearity making it a good option for the estimation of surface $NO_2$ concentrations from satellite observations. The trainable

100   parameters in a random forest model are the input covariates and the thresholds at which they are split. The main hyperparameters in the random forest model are the number of trees in the random forest, the loss function or evaluation criterion used to





**Table 1.** Hyperparameters tested and selected for the training of the machine learning algorithm.

| Hyperparameter | Range | Selected Value |
|---|---|---|
| max_depth | 1-100 | 16 |
| n_estimators | 1-100 | 30 |
| min_samples_split | 1-20 | 5 |
| min_samples_leaf | 1-10 | 2 |

assess the improvement training makes to the regression, the maximum number of splits a tree can have, and the maximum depth of the trees in the forest.

Since the method can be prone to over-fitting, it is important to select the model hyperparameters carefully, in order to obtain the best performance without overfitting the training data. Overfitting occurs when the model accurately predicts the training data but performs poorly on new data that was unseen during training. We split the data randomly into 90 % training and 10 % testing data to find the optimal hyperparameters. We trained random forests on the 90% of the data multiple times, iterating through a range of values for the hyperparamters. The selected hyperparameters are the ones that maximize the average $R^2$ on both the training dataset *and* the unseen test data and ensuring that the difference between the testing and training $R^2$ is smaller than 0.1. The final hyperparamters we used are presented in table 1, further details on the impact of the hyperparameters on the correlation of the testing and training dataset can be found in the appendix (Figure A1).

After hyperparameter tuning using the 10-fold cross validation method, we checked the performance of the model by splitting the available data into a training set encompassing the first 3 years (2019-2021) of data and a test set using only the 2022 data. We trained a random forest model using the best hyperparameters obtained above on the training set without letting the training see the test set. The spread of the $R^2$ between the training and test sets is then an indication of how much overfitting is happening. This training gave an $R^2_{test} = 0.78$ on the unseen 2022 data, while the average $R^2_{training} = 0.84$ for the 2019-2021 data, indicating a low degree of overfitting. This gives confidence that the model is not simply regurgitating the training data and can be safely used for predictions in areas and time periods away from the training dataset.

Finally, for the random forest predictions presented here, we used the same hyperparmaters and trained a model on all 4 years of available data to provide the best possible random forest model.

## 2.4 Input Parameters

To obtain the random forest fitting function, a training and test dataset needs to be established that consists of input parameters X and one output parameter Y. In our case the surface $NO_2$ measurements is the output parameter Y. For our final function we use the input parameters as listed in Table 2. Other parameters, including the ERA5 meteorological variables of 2 m surface temperature, total precipitation, relative humidity and the 2m dew point temperature, as well as the CO emissions have been tested in our random forest fitting. However, since none of these had a significant impact on the correlation (less than 0.005) they were consequently removed to avoid over-fitting and to reduce the computational burden. Additionally, we tested including





**Table 2.** Input parameters used for the training of the machine learning algorithm.

| Dataset | Parameter | Frequency | Source |
|---|---|---|---|
| $NO_2$ surface observations | output/training | daily/mid-day | AQS and NAPS |
| TROPOMI | $NO_2$ tropospheric columns | daily/mid-day | Copernicus |
| Meteorology | 10m winds (zonal and meridional), vertical velocity, surface pressure, boundary layer height (BLH) | daily/mid-day | ERA5 |
| Emissions | NO, $NO_2$, $SO_2$, $NH_3$ | daily/mid-day | GEM-MACH |
| Topography | Elevation | constant | |
| Landuse | IGBP landuse classification | constant | MODIS MCD12Q1.061 |
| population density | Gridded Population of the World (GPW) dataset (2020 projection) | constant | GPW Doxsey-Whitfield et al. (2015) |
| Temporal information | month | - | |

the location (longitude and latitude) of the stations, and while this overall improved the correlation, flaws started to appear when looking at maps containing locations that were not included in the training. Further details are discussed in Sect. 3.3.

### 2.4.1 ERA5 meteorology

Meteorology is an important component for an accurate prediction of surface concentrations. For example, the boundary layer height can determine the amount of $NO_2$ near the surface relative to the total column, as well as wind speed and direction can impact the column to surface concentration ratio. For this reason we included 10 m winds, vertical velocity, surface pressure, and boundary layer height, from the European Centre for Medium-Range Weather Forecasts (ECMWF) ERA5 reanalysis Dee et al. (2011) which is available hourly on 0.25×0.25 degrees resolution. Additionally, we tested using ERA5's 2-m temperature, the 2-m dew point temperature, total precipitation, and relative humidity, but found that it did not impact the correlation and had little effect on the random forest model. The ERA5 reanalysis data are interpolated to the nearest grid cell of the station/TROPOMI observation location. The meteorological parameters are averaged for the same hours as the insitu data (13 to 15 local time) for the training dataset and the closest hour is used for the TROPOMI L2 estimated surface observations.

### 2.4.2 Emission inventory

Including emissions inventory in the training of the model can help to pin-point location of elevated surface $NO_2$ concentrations thus we included the emissions of NO, $NO_2$, $SO_2$, and $NH_3$ in the machine learning algorithm. We use the same emission inventory as utilized in the Environment and Climate Change Canada's (ECCC's) operational regional air quality model Global Environmental Multi-scale - Modelling Air quality and CHemistry (GEM-MACH). These emissions are on a 10km×10km





resolution in Canada and the US and vary by hour, day of the week and month. The operational forecast makes use of 2013

emissions information and provides updated projections of emissions for 2023 in the US and Mexico and 2020 in Canada Zhang

et al. (2018). The emissions used in the model are processed using the Sparse Matrix Operator Kernel Emissions (SMOKE)

Coats (1996). For each measurement station, or TROPOMI location, the emissions of the closest grid box are used for the

coincident time.

## 2.5  GEM-MACH model

Output from the GEM-MACH model is used in two ways in this study: (1) for the traditional scaling method, and (2) to provide

surface concentrations for synthetic "stations" in remote areas (more details in Sect. 3.2). The operational version of the model

Moran et al. (2010); Pendlebury et al. (2018) has a $10 \times 10\,\text{km}^2$ grid cell size for North American domain, a 2-size bin aerosol

size distribution and 42 trace gases and eight particle species. GEM-MACH provides hourly output for a North American mod-

eling domain with an internal "physics" time step of 7.5 min. The chemical components of GEM-MACH reside as a subroutine

package within the model's meteorological physics model, the latter a component of the Global Environmental Multiscale

(GEM) weather forecast model Côté et al. (1998); Girard et al. (2014). The operational model run is "initialized", meaning the

meteorological parameters are replaced by analysis every 12 hours, at 00 and 12 UTC. Further details on GEM-MACH can be

found in, Makar et al. (2015b, a) and Akingunola et al. (2018).


## 2.6  Traditional scaling method

In previous studies, e.g. Lamsal et al. (2008); McLinden et al. (2014); Kharol et al. (2015); Griffin et al. (2019); Cooper

et al. (2020), surface concentrations from satellite observations are estimated by scaling the satellite column measurements

($\text{VCD}_{sat}$) by the model ratio of the surface concentration ($\text{C}_{model}$) and tropospheric columns ($\text{VCD}_{model}$) at the coincident

time and location:

$$C_{sat} = VCD_{sat} \times \frac{C_{model}}{VCD_{model}} \tag{1}$$

Because the GEM-MACH operational model currently does not contain free tropospheric emissions such as from aircraft

or lightning, we add a monthly mean from GEOS-Chem as the free tropospheric VCD to the GEM-MACH VCDs (further

details in Griffin et al. (2019)). These free tropospheric VCDs from GEM-MACH are on the order of $1\text{e}14\,\text{molec/cm}^2$ (as a

comparison, VCDs in polluted areas are on the order of $1\text{e}16\,\text{molec/cm}^2$).





## 3 Results

### 3.1 Validation and importance of parameters

Similar to previous studies, e.g. Long et al. (2022) we use 2022 to to assess the performance of the random forest algorithm to
predict the surface concentration, and used the 2018-2021 dataset for training, where $R^2_{training}$ = 0.77 and $R^2_{2022}$ = 0.77. The
result is shown in Figure 2a for the random forest approach and Figure 2d shows the same dataset using the traditional scaling
approach. Each point represents a single overpass pixel (mid-day) and location. The random forest approach shows significant
improvement ($R^2 = 0.79$) compared to the traditional ($R^2 = 0.38$) approach. Estimating annual or monthly means will increase
the correlation further. Monthly means are shown in Figures 2b and e for the random forest and traditional scaling, respectively.
Averaging the datasets increases the $R^2$ for both the machine learning ($R^2$=0.89) and the traditional scaling, however, the low
bias in the traditional scaling remains and the $R^2$ only reaches 0.5. The annual means for each station are shown in Figures
2c and f. While the traditional approach predicts much lower values, the random forest surface concentrations are close to the
1-to-1 line.

Similarly it is important to assess the importance of each input parameter used by the random forest function. The results
of the feature importance are shown in Figure 3 showing that the TROPOMI tropospheric column measurements are the most
important parameter for the prediction of the surface $NO_2$ concentrations. This is a feature that we explicitly wanted to see
in the random forest prediction, as this means the random forest function is primarily driven by the satellite measurements of
$NO_2$.

### 3.2 Limitation and improvements in remote areas

As mentioned briefly, we filled in $NO_2$ surface concentrations from the GEM-MACH model in remote areas for our training
dataset. Ground stations from NAPS and AQS are typically installed in urban or populated areas. This can be problematic for
machine learning algorithms, as the random forest predictor is only representative of its training data, which results in remote
areas much higher surface concentrations are computed than there actually are, if this is not taken into account. Including
model output in remote areas helps the random forest predictor to be able to predict low concentrations. To create the model
artificial "station" points we first created a regular grid of 1° in latitude between 60 and 40°N by 3° in longitude between 126
and 60°W. Then any points with a population density higher than 0 (utilizing the GPW dataset) were removed, leaving 290
remote locations (as shown in Figure 1) that are underrepresented by the NAPS and AQS datasets. This is roughly the same
amount as the NAPS stations and in total accounts for just under one third of the entire dataset used for the training of the
random forest predictor. This number was determined to be reasonable as this is less than the number of measurement stations
but a large enough collection to make an impact on the prediction in remote areas.

 An example of the impact is shown in Figure 4 for a remote CAPMoN (Canadian Air and Precipitation Monitoring Network)
measurement station located at Pinehouse lake in northern Saskatchewan (55.51°N, 106.72°W). This station is not part of
NAPS and thus was not part of the training dataset offering an excellent opportunity to evaluate estimated $NO_2$ surface con-





centrations in a remote area, note that the measurements have a detection limit of 0.09 ppbv. Figure 4a shows the measurement data in the 2019 summer (black dots). The estimated NO$_2$ concentrations are shown as blue squares when only NAPS and AQS stations are used for the training of the model, and as turquoise diamonds when the additional model "stations" are included in the training data. The results using the traditional scaling method are shown as purple points. Figure 4b shows estimated versus the measured surface concentrations for coincident data points. The current version utilizing the random forest predictor with

NAPS, AQS and model surface concentrations as training data is the closest to the observations. This example highlights the high bias when only station measurements are used as the random forest predictor is unable to predict low concentrations due to a lack of low concentrations in the training dataset. Using the additional surface concentrations from the GEM-MACH model helps the prediction in remote areas. The traditional method seems to be reasonable in remote areas but tends to underestimate the measurements slightly and is not as good as the estimated surface concentrations using the current version of the random

forest model.

### 3.3 Limitation and improvements to create maps

Figure 5 shows the estimated surface concentrations for the month of May 2023. Where the model was trained with data from 2018-2022. Figure 5a shows the current (best) version using AQS, NAPS and model for the training and eliminating the location with input parameters as listed in Table 2. For comparison the model scaled surface concentrations are shown

in Figure 5b, and are generally much lower than the random forest estimated values in urban areas. Figure 5c highlights the issue of over-predicting surface concentrations in remote areas when only NAPS and AQS are used for the training of the random forest predictor. Remote areas can show monthly average NO$_2$ surface concentrations of approximately 1 ppbv, but as high as 5 ppbv (e.g. over Greenland) which is not realistic. The next panel (Figure 5d) shows the impact of using latitude and longitude in the training input parameters, the random forest predictor tries to interpolate between the locations resulting

in sudden gradients across the map that are not realistic features. However, for specific measurement stations the correlation between the estimated and observed NO$_2$ is much improved when using location information as input parameters, it depends on the purpose of the prediction: if the purpose is to fill in data gaps at specific stations better results are archived when latitudes and longitudes are included in the input parameters and the random forest is trained for the specific location. As can be seen on the map, it is not advised to use location information for predicting locations not included in the training dataset as realistic

maps cannot be estimated as the random forest model tries to interpolate between locations. Therefore, for predicting surface concentrations where no station measurements are available, locations data (longitude and latitude) should not be included in the training dataset of the random forest model.

### 3.4 Application examples and comparison to the traditional method

Lastly, this version of the NO$_2$ surface concentrations is a satellite level 2 product meaning they are derived on individual

TROPOMI pixels. An example of a single satellite overpass is illustrated in Figure 6 highlighting the improvement of the random forest predictor over the traditional scaling method. The original TROPOMI NO$_2$ VCDs are shown in Figure 6a over the Greater Toronto Area (GTA) on May 23, 2023 (a clear-sky day), and no 2023 data has been used for the training of the





random forest model. The NO$_2$ surface concentrations using the traditional scaling method and the random forest predictor are shown in Figure 6 b and c, respectively. The coincident measurements from the NAPS stations in this area are included as points

(using the same color scale). This highlights the discrepancy of the traditional method that relies on the model profiles. The traditional scaling method shows largest enhancement (of approximately 6.5 ppbv) north of Toronto in this example. It appears much lower than the coincident NAPS measurements. The random forest estimated surface concentrations correlate well with the location of the TROPOMI VCD enhancements and the enhancement measured by the NAPS station. The observations and depict a realistic spatial picture of similar magnitude to the NAPS measurements. It should be noted that this was a clear-sky

day and cloudy days will result in significant gaps due to low quality satellite observations, not all days compare quite as well to the NAPS measurements.

## 4    Results and Future Implications

To summarize, the random forest predictor has demonstrated significant advantages over traditional method of model profile scaling in predicting surface concentrations of NO$_2$ from TROPOMI NO$_2$ observations.

Improving the prediction of surface concentrations through machine learning involves a comprehensive approach that includes careful consideration of data sources, input parameters, hyperparameter tuning, and most importantly rigorous validation and testing. This is crucial to ensure that the machine learning model can accurately reflect the complex dynamics of the atmosphere and make realistic predictions without over-fitting.

One of the key strengths of machine learning models is their excellent performance to fill in data measurement gaps, when

location data is included in training. This allows for accurate predictions of concentrations in different years and is particularly useful in scenarios where ground-based instruments fail, face measurement gaps, or are decommissioned. Including the location information, however, limits the ability to predict the surface concentrations in unknown locations and can create odd gradients as the predictor tries to interpolate between locations.

Another important consideration is that surface stations often under-represent remote areas as they are typically in urban

and polluted areas. This under-representation of remote areas can cause machine learning models such as the random forest predictor to over-predict concentrations when trained solely on available measurements. To overcome this limitation we trained the random forest machine learning, with additional model surface concentrations from ECCC's operational air quality forecast model in remote areas. This significantly enhanced the predictability of surface NO$_2$ concentrations across Canada and the US, including remote regions.

It should further be noted that the accuracy of these predictions is inherently tied to the quality of the training data. For example, if the chemiluminescence-based analyzers suffer from overestimation of NO$_2$ due to interference from PAN and nitric acid, the random forest estimated values will also overestimate the surface concentrations. The new satellite-derived surface NO$_2$ concentrations are complimentary to surface station monitoring as it relies on TROPOMI NO$_2$ VCD measurements to help fill in measurement gaps or areas that are currently unmonitored, which performs better than relying on CTM model out-

put alone. The estimated values are typically better than using the model scaling method, but are still not flawless. Outliers are



often challenging to predict, and the estimated surface concentrations show typically less spread and variation compared with the actual measurements. As an example, in 2022 there are 3 exceedances of $NO_2$ surface concentrations greater than 60 ppb for coincident dataset with TROPOMI, while the random forest predicted $NO_2$ surface concentrations were around 30-40 ppb. On-going validation with ground-based data remains essential. Furthermore, the models currently predict only mid-day surface
concentrations, at the time of the TROPOMI satellite overpass.

In the near future, the random forest predictor can potentially be applied to observations from the geostationary Tropospheric Emissions: Monitoring of Pollution (TEMPO) satellite Zoogman et al. (2017), which has the potential to create hourly daytime $NO_2$ surface concentration maps for North America.

*Code availability.*  Scripts used to create the figures in this manuscript and tune the random forest model can be found on github: https://github.com/DGriffin-eccc/NO2surface.

*Data availability.*  TROPOMI data can be downloaded from https://s5phub.copernicus.eu. Surface data from AQS (In the US) are available to download from https://www.epa.gov/outdoor-air-quality-data and from NAPS (in Canada) from https://www.canada.ca/en/environment-climate-change/services/air-pollution/monitoring-networks-data/national-air-pollution-program.html. TROPOMI surface concentrations using model scal-
ing and random forest as presented in this study can be found here: https://hpfx.collab.science.gc.ca/~deg001/surfaceNO2.





**Figure 2.** Validation for the 2022 dataset showing a), b) and C) for the random forest approach for daily, monthly and annual concentrations.

The model scaling method for the same data points is shown in d), e) and f) for daily, monthly, and annual concentrations.





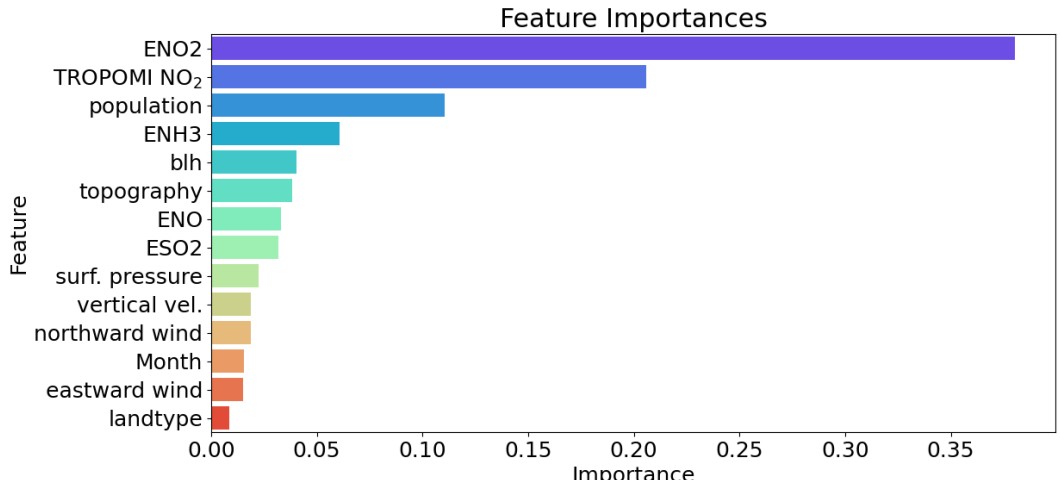

**Figure 3.** Importance of the input parameters for the random forest predictor.

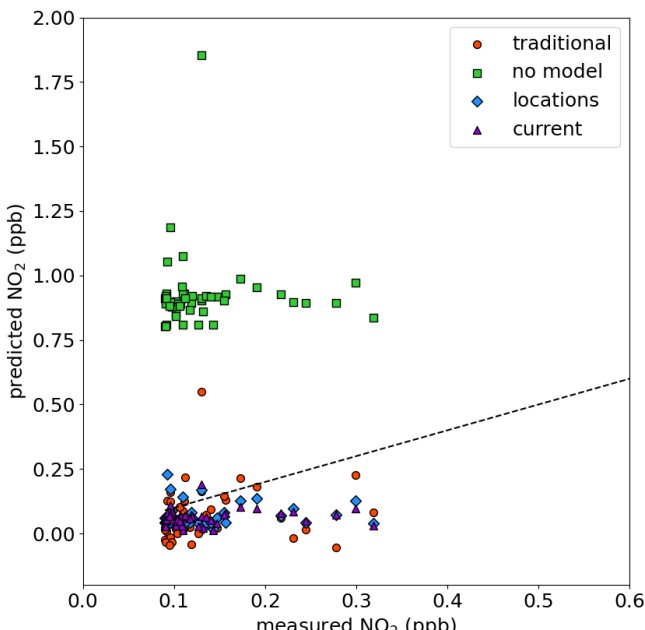

**Figure 4.** Comparison between measured and TROPOMI estimated $NO_2$ surface concentrations in the remote area Pinehouse lake (55.51°N, 106.72°W). The figure shows the correlation between the measured and estimated $NO_2$ for coincident measurements. Note that much higher concentrations up to 3 ppb were measured by the CAPMoN instrument, but these are not coincident with the TROPOMI overpasses.

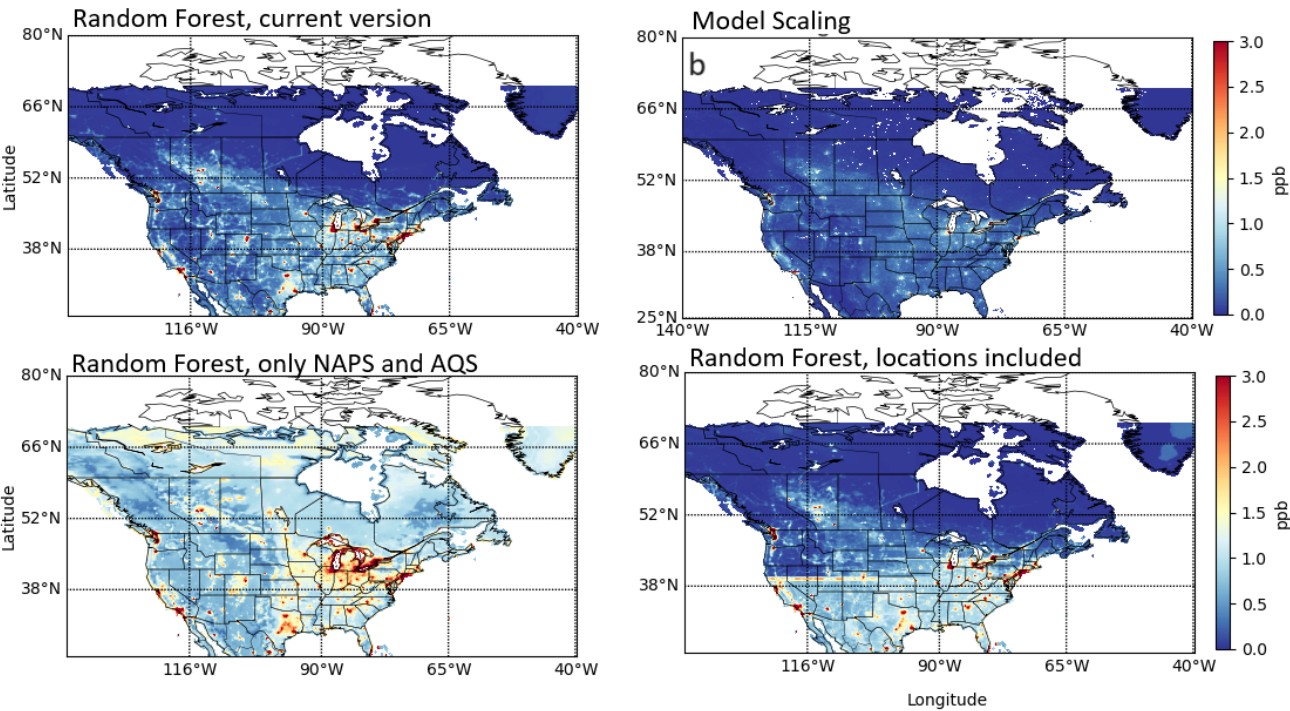

**Figure 5.** Monthly average of surface $NO_2$ concentrations for May 2023 over North America (binned on 0.1 with 0.2 oversampling). All figures have the same color scale. Panel a shows the current version of random forest estimated $NO_2$, panel b shows the model scaling $NO_2$ concentrations, panel c shows the random forest estimated $NO_2$ if only NAPS and AQS stations are used for the training, and panel d shows the random forest estimated $NO_2$ if locations (latitude and longitude) are included in the training data.



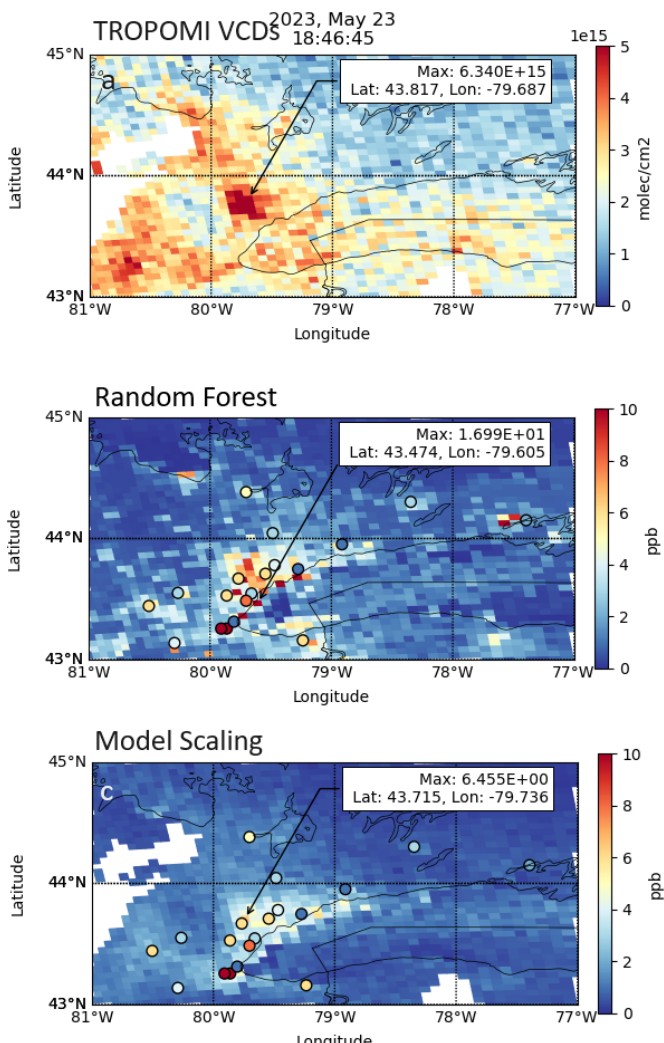

**Figure 6.** An example is shown a single overpass of the Toronto area on May 23, 2023, a clear-sky day. Panel a shows the TROPOMI VCDs, panel b shows the random forest estimated $NO_2$ concentrations, and panel c shows the model scaling $NO_2$ surface concentrations. Highlighted are the points with the greatest enhancement. The points shown in panel b and c are the coincident NAPS measurements on the same color scheme. For random forest training observations from 2018-2022 were considered (2023 is not included).





## Appendix A: Tuning of the random forest model

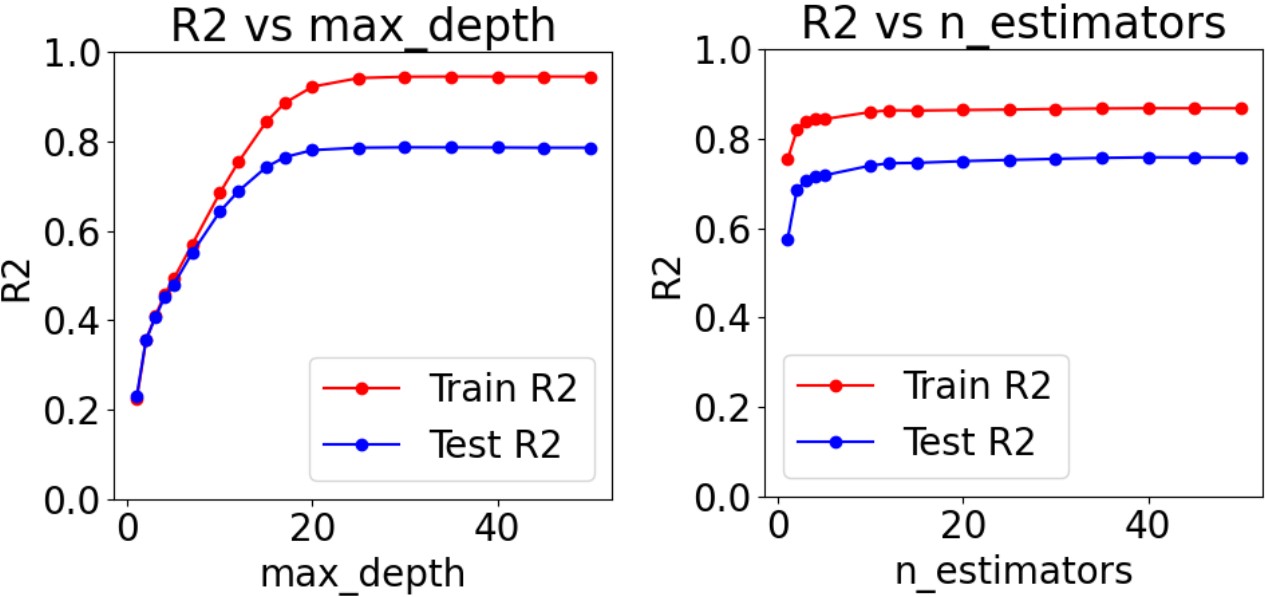

**Figure A1.** Figure illustrating on how the optimal parameters for the random model were found. The max_depth and n_estimators were the most important parameters that needed tuning. The ideal point is found where the correlation of the training data is similar to the correlation of the test dataset. It is a sign of over-fitting if the correlation coefficient is much lower for the test dataset than for the training dataset, this is most common for a too large max_depth. N_estimators has a significant impact on speed.

*Author contributions.* DG prepared the article with contributions from all co-authors. DG and CH prepared the data analysis and machine learning algorithms. CM, SKK, CL, CS, and MS helped develop the conceptual framework and methodology. SKK provided data. AF contributed to the data visualization. YY provided and prepared the CAPMoN data.

*Competing interests.* The authors declare no competing interests.

*Acknowledgements.* The authors acknowledge Environment and Climate Change Canada for the provision of nitrogen and sulphur species data from the Canadian Air and Precipitation Monitoring Network accessed from the Government of Canada Open Data Portal.



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
