# Peer review of "Development and validation of satellite-derived surface $NO_2$ estimates using machine learning versus traditional approaches in North America"

_EGUsphere, 2025_

## Referee Comment (RC1)

**Review comments for "Development and validation of satellite-derived surface NO₂ estimates using machine learning versus traditional approaches in North America"**

The study focuses on a current important topic of estimating spatial surface $NO_2$ using high resolution satellite instruments such as TROPOMI. Authors use an RF based ML approach to derive surface $NO_2$ and compare these results with the ones derived using the traditional scaling approach, focusing on region of North America. An interesting approach of additionally including model data as training targets to include under-represented remote areas seems to overcome the typical under-prediction issues of ML models in the background regions. This is clearly depicted in this study and is an interesting find that can help with accurate spatial predictions, not only over urban regions where stations are usually located, but also over rural and remote regions, highlighting the importance of integrating ML, satellite observation and model data. The findings of the study can be of interest to the scientific community working on air pollution.

While the manuscript is generally well-structured, scientifically sound, and includes relevant results and figures, there are a few minor issues to be addressed before it can be accepted for publication. In particular, some sentences would benefit from improved clarity and framing. I encourage the authors to carefully revise the manuscript, especially the sections highlighted below, to better communicate their findings. Furthermore, the manuscript is missing a more focused and detailed comparison between the ML-based and traditional approaches, which forms the main objective of the study.

Specific comments

1. Line 55: Please mention the target spatial resolution of your model somewhere in the methodology.
2. Line 64: The changes in versions of TROPOMI through the years results in some differences between the TROPOMI NO2 values for high pollution levels. Could you please explain why you choose this version when a harmonized reprocessed version is available?
3. Line 88: Figure 1 indicates the location of the model parameters as well. Why do you consider the model parameter points over ocean, if you are not predicting there? Will that not impact or introduce some noise into the model as the meteorological conditions over ocean are quite different than over land and hence these datasets can act as noise. Care must be taken to analyze any negative impact of this.
4. Line 97: Citation required for Random Forest
5. Line 100: The logic behind the 0.1 threshold needs a one-liner explanation or a citation

6. Line 123: Please mention a line or two on the data preparation part that includes how and to what resolution the input parameters were gridded or regridded to, since they all are in different spatial resolution? Were there any transformations applied on the datasets to deal with their general skewness? If yes, please mention.

7. In Table 2 caption, please indicate shortly what does the combination daily/mid-day mean. This helps a reader to gauge all the necessary information from the table, without having to search in the main text. Also, of relevance here is the spatial resolution of input parameters

8. Line 130: Why not ERA5-Land for meteorology such as wind and surface pressure which is available at higher spatial resolution than ERA5 and is more relatable to surface variations.

9. Figure 2: The color scale for density is missing and needs to be added

10. Line 185: How is the feature importance derived here? I suppose, in your case it's from the inbuilt feature importance function from random forest? Please mention this in the methods.

11. Line 194: Do you mean "which results in remote areas predicting/computing much higher concentrations than there actually are"? Please check this sentence in terms of phrasing it right

12. Line 205 and 208: Figure 4a and Figure 4b seem to be missing. There is only one figure 4. The black dots mentioned as part of Figure 4a are missing. Please check this result.

13. Line 210: Please check the logical flow of this sentence and correct the same. The result is not communicated clearly. Also, it would be good to specify the direction of bias in certain discussions, i.e, positive or negative. Would recommend modifying the sentence into something like "This example highlights the high positive bias when only station measurements are used as random forest predictor. The RF model is unable to predict low concentrations due to lack of low-concentration scenarios in the training data"

14. Line 210: Does this mean that RF without model inputs, will generally not work well in background locations? How about during low $NO_2$ periods such as summer over urban regions? ML models usually overestimate low pollution levels and the inclusion of GEM-MACH model in your case, perhaps helps with the overall improvement in accuracy during low pollution levels (even over urban regions) and can also be generalized and stated as such in the study. This will be an important contribution and consider mentioning it after verification.

15. Line 220: More comparative discussion between Fig 5a and Fig 5b should be included, as this is also the focus as per the title of the paper.

16. Line 225-226. Please highlight in the map by overlaying correlation values or something similar to indicate the data filling capabilities of the model when latitude longitude is included as inputs. Yes, we can see the weird patterns at some locations, but which locations indicate the gap filling capacity is not clear or evident. Currently there is no statistical proof or indicator to show the same.

Please consider adding a figure (may be a zoomed in version) or a table with correlation values to indicate this.

17. Line 240: What is the reason for less accuracy in the traditional approach here?
18. Line 243: Looks like authors missed including something after "The observations and". Please check this.
19. Line 269-270: The authors state that the RF-based current approach performs better than relying on CTM output alone. Where is this conclusion referred to in your result?
20. Line 272-273: Does the RF version without the model as input capture these exceedances? It would be interesting to verify if the inclusion of model data is lowering these peak predictions?

Looking forward to seeing the revised version of this manuscript with the above changes. I thank the authors for the manuscript and wish them a good luck.

---

## Author Comment (AC1)

**Response to reviewer1:**

We would like to thank Reviewer 1 for their comments and suggestions. We have addressed them in the new version of the manuscript. Our responses are highlighted below in red following the reviewer comments.

Review comments for "Development and validation of satellite-derived surface NO₂ estimates using machine learning versus traditional approaches in North America"

The study focuses on a current important topic of estimating spatial surface NO2 using high resolution satellite instruments such as TROPOMI. Authors use an RF based ML approach to derive surface NO2 and compare these results with the ones derived using the traditional scaling approach, focusing on region of North America. An interesting approach of additionally including model data as training targets to include underrepresented remote areas seems to overcome the typical under-prediction issues of ML models in the background regions. This is clearly depicted in this study and is an interesting find that can help with accurate spatial predictions, not only over urban regions where stations are usually located, but also over rural and remote regions, highlighting the importance of integrating ML, satellite observation and model data. The findings of the study can be of interest to the scientific community working on air pollution.

While the manuscript is generally well-structured, scientifically sound, and includes relevant results and figures, there are a few minor issues to be addressed before it can be accepted for publication. In particular, some sentences would benefit from improved clarity and framing. I encourage the authors to carefully revise the manuscript, especially the sections highlighted below, to better communicate their findings. Furthermore, the manuscript is missing a more focused and detailed comparison between the ML-based and traditional approaches, which forms the main objective of the study.

We revised the manuscript as detailed below. As suggested, we included a more detailed comparison between the traditional and ML approach in the manuscript. See details below.

**Specific comments**

- 1. Line 55: Please mention the target spatial resolution of your model somewhere in the methodology.
  - This is done on L2 data, so the resolution is equal to the resolution of TROPOMI (~5x3.5 km2), additionally we would like to highlight that averaging this dataset (similar as to TROPOMI VCDs are averaged would improve the surface concentrations and reduce the noise). We included the following sentence in the manuscript:
  - "In this study, we use machine learning models to obtain surface concentrations for each TROPOMI observation (L2) on the same spatial resolution as the satellite observations themselves."
- 2. Line 64: The changes in versions of TROPOMI through the years results in some differences between the TROPOMI NO2 values for high pollution levels. Could you please explain why you choose this version when a harmonized reprocessed version is available?
  Version 2 is the latest TROPOMI processing, we use a combination of reprocessed and OFFL (for newer data) which is the same version as RPRO.

- 3. Line 88: Figure 1 indicates the location of the model parameters as well. Why do you consider the model parameter points over ocean, if you are not predicting there? Will that not impact or introduce some noise into the model as the meteorological conditions over ocean are quite different than over land and hence these datasets can act as noise. Care must be taken to analyze any negative impact of this.
  - Thanks for pointing this out. Originally, we did not plan to remove the points over oceans, but we have limited ways to verify the random forest surface concentrations over water and were not certain if these RF surface concentrations represent the truth well. There is flexibility to keep the ocean points and evaluate at a later point.

We did a test removing the synthetic stations over the ocean, see figures below. As a result, there are not enough "synthetic" background stations, and the values in rural/northern areas are worse. We created similar figures as Fig. 2, 4, and 5. With points over ocean removed for the training model. In Fig. 2 no significant difference is obvious (this is for all NAPS and AQS stations), however looking at rural stations such as the CAPMoN Pinehouse Lake location, as well as the map, removing the points over ocean performs worse. Alternatively, more "synthetic stations" could be added over land likely resulting in similar results as the original random forest model. Creating these additional datasets for additional "synthetic stations" over land requires a lot of work effort with limited improvement of the end result.

**We included the following sentence in the manuscript: "While the surface concentrations are available for each TROPOMI observation, the learning—derived NO2 surface concentrations over water are not shown here because their validity could not be verified in the absence of**

monitoring stations.
[....] A similar amount of synthetic background stations as NAPS stations is needed for the machine learning model to improve in background areas. "

Observed NO2 (ppb)

Below are the figures synthetic ocean stations removed.

4. Line 97: Citation required for Random Forest
We included the following citation here: Breiman et al., 2001, Random Forests, Machine Leaning,
45, 5-32, https://doi.org/10.1023/A:1010933404324

5. Line 100: The logic behind the 0.1 threshold needs a one-liner explanation or a citation We changed the sentence in the manuscript to:

"The selected hyperparameters are the ones that maximize the average R2 on both the training dataset and the unseen test data while ensuring that the difference between the testing and training R2 is small (indicating that the model has not overfit the training data and generalizes well to unseen data). There is no generally accepted definition of "small" so we chose a threshold of 0.1. At this location in the hyperparameter space, the test R2 value is not very sensitive to small changes in any of the hyperparameters."

6. Line 123: Please mention a line or two on the data preparation part that includes how and to what resolution the input parameters were gridded or regridded to, since they all are in different spatial resolution? Were there any transformations applied on the datasets to deal with their general skewness? If yes, please mention.

We included the resolution of the dataset in Table 2.

- 7. In Table 2 caption, please indicate shortly what does the combination daily/midday mean. This helps a reader to gauge all the necessary information from the table, without having to search in the main text. Also, of relevance here is the spatial resolution of input parameters. These parameters change daily, technically all the time, but since TROPOMI only has one value per day at mid-day we take the mid-day point. While other parameters are constant, as indicated in the table. We included the following to make it clearer in the table caption: "Some parameters are constant while others change hourly, however since TROPOMI only observes at mid-day, only coincident points with the TROPOMI observations are selected and indicated by "daily/mid-day" in the Table below."
  - The table also includes now the spatial resolution of the input parameters.
- 8. Line 130: Why not ERA5-Land for meteorology such as wind and surface pressure which is available at higher spatial resolution than ERA5 and is more relatable to surface variations. We would like to thank reviewer 1 for this suggestion of using ERA land which is at a higher resolution than ERA5. This is a good idea and we will update to this dataset in our future studies. For this study, however, we originally did not intend to remove the points over water and thus ERA-Land would have been insufficient. At this point including ERA land instead of ERA5 will be a big work effort and we would have to start from the beginning again. We also obtain NO2 surface concentration information over water, but these were removed in this study because we did not have a way to verify its validity. Considering that the meteorological parameters such as the pressure and wind have very low importance (less than 0.05, see Fig. 3), we do not believe that including ERA5 land will have any significant impact on the machine learning model results. We included the following in the manuscript:
  - "ERA5 was chosen over ERA-Land (which has the advantage of a higher spatial resolution), because it is available for each TROPOMI observation, and surface concentrations over water could be obtained."
- 9. Figure 2: The color scale for density is missing and needs to be added Thanks for pointing this out. Figure 2 has been changed as suggested.
- 10. Line 185: How is the feature importance derived here? I suppose, in your case it's from the inbuilt feature importance function from random forest? Please mention this in the methods. Yes, there is a built-in function that can be used for this. We have included the following in the manuscript: "The results of the feature importance (obtained through an in-built function of sklearn "feature\_importances\_") are shown in Figure 3 showing..."
- 11. Line 194: Do you mean "which results in remote areas predicting/computing much higher concentrations than there actually are"? Please check this sentence in terms of phrasing it right We have corrected this sentence as suggested and included "computing".
- 12. Line 205 and 208: Figure 4a and Figure 4b seem to be missing. There is only one figure 4. The black dots mentioned as part of Figure 4a are missing. Please check this result.

Thanks for pointing this out, we included Figure 4 a, somehow this went missing. Please find below the updated figure, which has also been updated in the manuscript:

13. Line 210: Please check the logical flow of this sentence and correct the same. The result is not communicated clearly. Also, it would be good to specify the direction of bias in certain discussions, i.e, positive or negative. Would recommend modifying the sentence into something like "This example highlights the high positive bias when only station measurements are used as random forest predictor. The RF model is unable to predict low concentrations due to lack of low concentration scenarios in the training data"

As suggested, we revised the sentence in the manuscript to:

"This example highlights the high positive bias when only station measurements are used as random forest predictor. The RF model is unable to predict low concentrations due to lack of low concentration scenarios in the training data."

14. Line 210: Does this mean that RF without model inputs, will generally not work well in background locations? How about during low NO2 periods such as summer over urban regions? ML models usually overestimate low pollution levels and the inclusion of GEM-MACH model in your case, perhaps helps with the overall improvement in accuracy during low pollution levels (even over urban regions) and can also be generalized and stated as such in the study. This will be an important contribution and consider mentioning it after verification.

Yes, the RF does not work well (with just NAPS and AQS stations) in rural areas and background locations, as can be seen in Figs. 4 and 5. Typically the ML models will tend towards mean values and does not predict outliers well. We included the following in the manuscript:

"Generally, the random forest model does not do well predicting extreme concentrations (high and low surface concentrations). Using the additional surface concentrations from the GEM-MACH model helps the prediction in remote areas where surface concentrations are typically low (see further discussion in the next section and Fig. 6). [...] Furthermore, including these synthetic rural stations also helps improve the overall accuracy during low pollution levels, see Fig. 5, shown is the frequency (illustrated in 5ppb bins) of surface concentrations (compared to measurements "Insitu Data"), when synthetic stations are not included the surface concentrations are often overestimated for surface concentrations less than 10

ppb, significant improvement is shown when the synthetic GEM-MACH stations are included otherwise the RF surface concentrations are too high. The traditional method appears to have more frequent points on the lower end (0-5 ppb) compared to the actual measurements."

15. Line 220: More comparative discussion between Fig 5a and Fig 5b should be included, as this is also the focus as per the title of the paper.

We included another figure showing the frequency of occurrences, including the traditional method, see comment above. We added the following to the manuscript:

"The traditional method appears to have more frequent points on the lower end (0-5 ppb) compared to the actual measurements, showing that the traditional scaling method tends to underpredict the true surface concentrations."

Also, as suggested we elaborated the discussion on Fig 5 a and b, and included the following in the manuscript:

"...lower than the random forest estimated values, especially in sub-urban and urban areas. Compared to the station measurements the traditional method is typically under-predicting and the current version of the random forest model shows are more similar pattern to the actual measurements (see Fig. 5)."

16. Line 225-226. Please highlight in the map by overlaying correlation values or something similar to indicate the data filling capabilities of the model when latitude longitude is included as inputs. Yes, we can see the weird patterns at some locations, but which locations indicate the gap filling capacity is not clear or evident. Currently there is no statistical proof or indicator to show the same. Please consider adding a figure (may be a zoomed in version) or a table with correlation values to indicate this.

We added more details and included a Figure similar to Fig. 2 but also using the location in the training of the RF model:

"However, for specific measurement stations the correlation between the estimated and observed NO2 is much improved when using location information as input parameters the correlation is better when using the location information in the RF model R2=0.8, see Fig. A1, whereas the current version of the

**model only achieves R2=0.77)..."**

17. Line 240: What is the reason for less accuracy in the traditional approach here?

Relying on model profiles for the scaling, which relies on boundary layer height, emissions, winds... We included more details in the manuscript:

"This highlights the discrepancy of the traditional method that relies heavily on the model profiles and surface concentrations that are used for the scaling. The model profiles and surface concentrations are influenced by the location and magnitude of emissions in the GEM-MACH model, the accuracy of the winds and the boundary layer heights."

18. Line 243: Looks like authors missed including something after "The observations and". Please check this.

We revised the sentence and removed "and".

19. Line 269-270: The authors state that the RF-based current approach performs better than relying on CTM output alone. Where is this conclusion referred to in your result?

This was a typo, we meant to say:

"... which performs better than the traditional scaling method."

20. Line 272-273: Does the RF version without the model as input capture these exceedances? It would be interesting to verify if the inclusion of model data is lowering these peak predictions?

No even when the model "stations" are excluded the exceedances are still not captured. There are just not enough exceedances in the training dataset. We included a figure in the appendix showing the random forest (same as Fig. 2) without the "model stations". We also included the following sentence in the manuscript:

"This is the case for all RF tests performed in this study."

Looking forward to seeing the revised version of this manuscript with the above changes. I thank the authors for the manuscript and wish them a good luck.

Thank you!

---

## Author Comment (AC2)

**Responses to Reviewer2:**

We would like to thank Fei Liu for her review of our manuscript. We have addressed her comments and suggestions and updated the manuscript accordingly. Our responses are indicated in red below the comments.

The authors develop a machine learning algorithm to infer surface NO2. The manuscript is well-written, and the results appear robust. I recommend the paper for publication after minor revisions.

**General comments:**

1. The authors state, "Currently, to our knowledge, there is little machine learning done to derive surface concentrations in less populated areas such as Canada." While this points out the motivation of the work, it would be helpful to expand on why this is significant. What differences between more populated regions (e.g., China, Germany) and less populated regions (e.g., Canada) would justify the need for this study? This additional context could better motivate the work.

We expanded the text in the manuscript further to justify the work and the importance of NO2 in background areas:

"However, to our knowledge, few studies have focused on applying such methods in less populated regions such as Canada. This gap is significant because the conditions in Canada differ significantly from those in densely populated regions: surface monitoring networks are sparser, emission sources are different compared to urban areas, and the limited spatial extent of measurement networks in northern Canada contribute to the challenge. These differences highlight the need to adapt and evaluate ML approaches under these conditions."

2. Section 2.4.2. I found it surprising that, in addition to NOx emissions, SO2 and NH3 emissions were included as predictors. Have you tested the sensitivity of your model to these additional species? It would be worthwhile to elaborate on why these variables are expected to play a role in predicting surface NO2. In Figure 3, I observe that NO2 emissions are far more significant compared to other parameters, including TROPOMI NO2. This observation seems inconsistent with the conclusion that "This is a feature that we explicitly wanted to see in the random forest prediction, as this means the random forest function is primarily driven by the satellite measurements of NO2." Clarifying this potential discrepancy would strengthen the argument.

Figure 3 includes the feature importance of all parameters used, including NH3 and SO2. We included the following sentence into the manuscript to clarify this more:

"While the NO2 and NO emissions are directly related to the NO2 surface concentration, other pollutants can be helpful in the machine learning model as well. SO2 emissions are typically an indicator of industrial activity, such as refineries potentially affecting the surface NO2 concentrations differently compared to urban emissions. While NH3 emissions are typical indicators of agricultural and some industrial activities nearby, NH3 can also impact the

formation NOx (Pai et al., 2021). The importance of the various parameters is discussed in Sect. 3 (Fig. 3)"

With regards to the discrepancy in the text, we changed the sentence to:

"...that the TROPOMI tropospheric column measurements are among the most important parameters for the prediction of the surface NO2 concentrations."

3. The authors mention that NO2 surface concentrations from the GEM-MACH model were used to augment the training dataset in remote areas. Would assigning greater weight to rural monitoring stations achieve a similar effect? If this has not been tested, it would be worthwhile to evaluate whether such an approach could improve the model's predictions for less populated regions.

We would like to thank Fei for the suggestion, we tested out weighting the background stations (stations with 0 population around) in two ways, weighting them 5 times more than all other stations and 10 times more, respectively. We created similar figures as Fig. 2,4, and 5. While Fig. 2 suggests very similar results as using the synthetic station data for the training, but looking at rural areas on the map or compared to the CAPMON rural site it performs much worse. The results are shown below.

This is an interesting analysis and worthwhile to elaborate on this in the manuscript. We included a brief discussion on this into the manuscript:

"Additionally, we tested weighting stations that are in non populated areas five and ten times more than other stations. This method did not work as well as including synthetic stations, the NO2 surface concentrations in northern Canada were still too high even when rural stations are weighted more in the random forest training (see Fig. A4)."

**Specific comments:**

1. line 170. 1e14 shall be superscript.

We have changed it accordingly to: "...order of  $10^{14}$  molec/cm2 (as a comparison, VCDs in polluted areas are on the order of  $10^{16}$  molec/cm2)."

---

## Referee Report (RR1)

Review Comments – Round 2

I thank the authors for clarifying the review questions and making the recommended changes in the manuscript. I would like to point out one aspect that still needs improvement for an article to be accepted in a scientific journal – presentation of figures. Please see below for recommendations on this.

1. Figure 2 – Labels a,b,c,d,e,f are missing. The last column elements should align with the subplot of size of other figures in the row. I recommend decreasing the size of the subplots in last column to match the others.
2. Figure 3 and 5 – Minor but please make sure the sub/superscripts for labels such as $NO_2$, $NH_3$ is properly done throughout.
3. Figure 7 – Labeling of panels missing. Also please fix the date title on the top so that it doesn't overlap with the title of the panel a

---

## Author Response (AR2)

We thank the reviewers for taking a second look at our manuscript.

Based on the suggestion, we have made the labels between the figures more consistent and changed some of the labels to be easier to follow.